# Pharmacogenetics and Adverse Events in the Use of Fluoropyrimidine in a Cohort of Cancer Patients on Standard of Care Treatment in Zimbabwe

**DOI:** 10.3390/jpm13040588

**Published:** 2023-03-28

**Authors:** Boluwatife Lawrence Afolabi, Tinashe Mazhindu, Chikwambi Zedias, Margaret Borok, Ntokozo Ndlovu, Collen Masimirembwa

**Affiliations:** 1African Institute of Biomedical Science and Technology, Harare P.O. Box 2294, Zimbabwe; bolu.afolabi@gmail.com (B.L.A.);; 2Department of Biotechnology, School of Health Sciences, Chinhoyi University of Technology, Chinhoyi Private Bag 7724, Zimbabwe; 3Department of Oncology, Faculty of Medicine and Health Sciences, University of Zimbabwe, Harare P.O. Box 2294, Zimbabwe

**Keywords:** adverse drug reactions, genetic polymorphism, cancer, capecitabine, fluorouracil, precision medicine

## Abstract

Fluoropyrimidines are commonly used in the treatment of colorectal cancer. They are, however, associated with adverse events (AEs), of which gastrointestinal, myelosuppression and palmar-plantar erythrodysesthesia are the most common. Clinical guidelines are used for fluoropyrimidine dosing based on dihydropyrimidine dehydrogenase (*DPYD*) genetic polymorphism and have been shown to reduce these AEs in patients of European ancestry. This study aimed to evaluate, for the first time, the clinical applicability of these guidelines in a cohort of cancer patients on fluoropyrimidine standard of care treatment in Zimbabwe. DNA was extracted from whole blood and used for *DPYD* genotyping. Adverse events were monitored for six months using the Common Terminology Criteria for AEs (CTCAE) v.5.0. None of the 150 genotyped patients was a carrier of any of the pathogenic variants (*DPYD**2A, *DPYD**13, rs67376798, or rs75017182). However, severe AEs were high (36%) compared to those reported in the literature from other populations. There was a statistically significant association between BSA (*p* = 0.0074) and BMI (*p* = 0.0001) with severe global AEs. This study has shown the absence of the currently known actionable *DPYD* variants in the Zimbabwean cancer patient cohort. Therefore, the current pathogenic variants in the guidelines might not be feasible for all populations hence the call for modification of the current *DPYD* guidelines to include minority populations for the benefit of all diverse patients.

## 1. Introduction

Fluoropyrimidines (fluorouracil and capecitabine) are components of the standard therapy for a variety of malignancies, including colorectal, pancreatic, gastric, breast, cervical, and head & neck cancer [1,2]. Different mechanisms of fluoropyrimidines activation into cytotoxic nucleotides have been described, one of them is the conversion of fluoropyrimidine to 5-fluoro-deoxyuridine monophosphate (FdUMP) that competitively inhibits the enzyme thymidylate synthase (TS), thereby creating a thymine deficiency and results in the inhibition of deoxyribonucleic acid (DNA) synthesis and cytotoxicity [3]. Studies have shown the efficacy of fluoropyrimidines in different cancer types, particularly in colorectal cancer patients, resulting in significant tumor reduction and growth [3,4]. However, fluoropyrimidines do not only affect cancer cells but also healthy cells, leading to dose-dependent toxicities [5,6]. Major toxicity from fluoropyrimidine treatment primarily reflects excessive cell death in healthy tissue with rapidly dividing cells, such as haematopoietic cells in the bone marrow and epithelial cells of mucous membranes [2,7]. The dihydropyrimidine dehydrogenase enzyme (DPD), encoded by the gene *DPYD,* plays a key role in the metabolism of fluorouracil. Genetic polymorphisms in the *DPYD* gene have been shown to be potentially responsible for lethal toxicity after fluoropyrimidine-based chemotherapy. Over 160 single nucleotide polymorphisms have been annotated for the gene [5,8]. Studies including in vitro DPD enzyme activity and in silico functionality of DPD activity have shown the non-functionality of *DPYD**2A and *13, and also 50% reduced function of c.2846A>T and Hap B3 *DPYD* [9,10,11].

Based on the treatment regimen, particularly on the drugs combined with fluoropyrimidine, 10–40% of patients experience severe, and in rare cases (0.2–0.5%), lethal fluoropyrimidine-related toxicity in their early chemotherapy cycles [7]. Effective and standardised toxicity reporting on fluoropyrimidine’s adverse events has helped in drug safety research and the adoption of clinical guidelines to prevent adverse drug reactions in the European population [9,12]. For example, fluoropyrimidine adverse events studies in European countries aimed toreduce observed toxicity based on the clinical recommendation to a lower percentage of dose administered while maintaining its therapeutic effect [10,11]. Some of the commonly reported toxicities attributable to fluoropyrimidines are gastrointestinal (diarrhoea, nausea, and vomiting), myelosuppression (neutropenia, leukopenia, thrombocytopenia, and anaemia) and palmar-plantar erythrodysesthesia (hand-foot syndrome) [12]. 

Pharmacogenomics (PGx) has contributed to personalised medicine, which seeks to understand each individual’s genetic composition to optimise drug therapy [13,14,15]. It is increasingly being implemented in the clinical setting to stratify the patient population and alleviate the burden of adverse drug reactions (ADR) through individualised therapy [16,17,18]. However, several barriers are still slowing down its widespread clinical implementation. One of the major barriers in African countries is the limited number of concise reports on adverse drug events both on admission and during hospitalisation [15,16]. Pharmacogenetic guidelines such as the Clinical Pharmacogenetic Information Consortium (CPIC), Dutch Pharmacogenetics Working Group (DPWG), and French National Network (Réseau) of Pharmacogenetics (RNPGx) have been developed to support the use of pharmacogenetic information in clinical settings. There are slight differences in these guidelines for fluoropyrimidine use based on the functionality of DPD. For example, DPWG advises initiating fluorouracil or capecitabine in patients with decreased DPD enzyme activity with a starting dose of 50% [19]; the CPIC recommendation of 50% dose reduction is followed by dose titration based on the clinical judgement of the healthcare professional and therapeutic drug monitoring. At the same time RNPGx refers to dose and pharmaco-therapeutic recommendations of other guidelines. Overall, the approaches of DPWG, CPIC, and RNPGx are generally similar and were designed to help clinicians understand how available genetic test results should be used to optimise drug therapy. [13,17,19]. The guidelines that support the use of fluoropyrimidines are being developed based on genetic information from the western populations. However, the clinical implementation of pharmacogenetics has not yet been accepted worldwide [16]. It is increasingly being adopted in some African countries [13]. There is consensus to adapt pharmacogenetic guidelines for fluoropyrimidine dosages based on genotyping. Some reports have shown success in this approach, whilst others have shown that the approach only partially explains [17] or fails to explain [3] all fluoropyrimidines-related toxicity.

In Zimbabwe, fluoropyrimidines are mostly used in treating colorectal, breast, gastric, oesophagus, cervical, and head & neck cancer. Based on GLOBOCAN statistics in 2020, cervical cancer ranks first in Zimbabwe and second in Africa, breast cancer ranks second in Zimbabwe and first in Africa, and colorectal cancer ranks tenth in Zimbabwe and seventh in Africa [20,21]. However despite the prevalence of fluoropyrimidine usage, there are no data on the PGx and pharmacokinetics (PK) of fluoropyrimidines or the prevalence of fluoropyrimidine-related adverse events based on clinical evidence in people of African ancestry. Such knowledge could help evaluate the potential value of implementing PGx-guided fluoropyrimidine usage in Africa or help identify other variants that can be linked with fluoropyrimidine-related adverse events and used for African-specific dosing guidelines. This prospective observational study aimed at establishing the prevalence of fluoropyrimidine-associated adverse events, the frequency of currently known actionable *DPYD* genetic variants and the potential clinical applicability of current PGx guidelines in identifying at-risk patients in a cohort of cancer patients in Zimbabwe.

## 2. Materials and Methods

This study was a prospective observational study of histologically confirmed cancer patients prescribed fluoropyrimidine-based chemotherapy or eligible for fluoropyrimidine chemotherapy between 5 March 2022, and 5 September 2022. We obtained ethical approval from the Joint Research Ethics Committee (JREC) for the University of Zimbabwe Faculty of Medicine and Health Sciences and Parirenyatwa Group of Hospitals and the Medical Research Council of Zimbabwe (MRCZ) with approval number MRCZ/B/2303. Written informed consent was obtained from all the participants according to MRCZ guidelines. The study population consisted of adult cancer patients (≥18 years) who were scheduled to start or were already on fluoropyrimidine-based chemotherapy, either as a single agent or combined with other chemotherapeutic agents, radiotherapy, or both. The visual representation of the methodology is shown in Figure 1.

Whole blood was collected in a 4 mL EDTA tube from the participants who met the inclusion criteria. 200 µL of whole blood was processed for DNA extraction and subsequent genotyping procedures. The Kingfisher flex Magnetic Particle Processor (Thermo Fisher Scientific, Waltham, MA, USA) was used for DNA extraction, and the DNA was quantified using the Qubit 4 fluorometer (Thermo Fisher Scientific, Waltham, MA, USA). The *DPYD* pathogenic variants were determined using AiBST pharmacogenetic panel (GenoPharm^®^, Thermo Fisher Scientific) (*DPYD**2A, *DPYD**13, rs67376798, or rs75017182) according to the manufacturer’s instructions. Briefly, 6 µL of each 50 ng/µL DNA sample was mixed with 6 µL of the Open Array Genotyping Master mix (from Applied Biosystems San Diego, CA, USA) in a 96-well plate, vortexed, centrifuged and then transferred to a 384-well plate. A no-template control (NTC) was also included on the plate. The contents of the 384 well plate were transferred to an open array plate (33 nL per well) using the robotic arm of the Applied Biosystems Accufill System (Thermo Fisher Scientific, Marsiling Industrial Estaste, Singapore). The run was carried out on the QuantStudio 12K Flex RealTime PCR System (Thermo Fisher Scientific, Marsiling Industrial Estaste, Singapore) using the default PCR conditions pre-set by the manufacturer for an open array genotyping run. All calls were made at cycle 45 using the default quality value ≥ 0.95 to assign a genotype call.

Patients’ medical records from the registry department at Radiotherapy Centre, Parirenyatwa Group of Hospitals, Zimbabwe were assessed. The research physician and trained research scientist collected adverse events outcome data during daily ward rounds and patient clinic days. A simplified and computerized form was made based on feasible data that could be collected from the medical records and patients. The first section of the form recorded the demographic data and lifestyle of the patient. The second section recorded medical conditions, medication use, and other clinical variables; the third section of the form recorded toxicity/AEs data. Adverse events were only recorded if they occurred in patients on fluoropyrimidine during their treatment cycle and within the study period. Adverse events were defined using the definition used by the National Cancer Institute Common Terminology Criteria for Adverse Events (CTCAE) [22]. The severity grading of treatment-related hand-foot syndrome, haematological and gastrointestinal AEs was done according to CTCAEv.5.0. [22]. Adverse events from published data were retrospectively obtained from historical [23] and literature [24] cohorts from other populations for comparison with this cohort.

The major toxicity categories associated with fluoropyrimidine such as gastrointestinal (diarrhoea, nausea, and vomiting), myelosuppression (neutropenia, leukopenia, thrombocytopenia, and anaemia) and palmar-plantar erythrodysesthesia (hand-foot syndrome) were evaluated for association with the clinical predictor (body surface area, BSA and body mass index, BMI) of fluoropyrimidine AEs.

## 3. Results

### 3.1. Study Participants Characteristics

The flow diagram of the study cohort is shown in Figure 2 below. We recruited 150 patients and did genotypes for all. Patients on fluoropyrimidine (50) were followed up for AEs within the study period (6 months). Adverse events data were recorded for each patient on fluoropyrimidine from the day of their recruitment into the study. One hundred (100) patients were not on a fluoropyrimidine, although eligible for it and therefore were not followed up for AEs. The physician’s choice of chemotherapy was based on patient management and standard of care treatment.

The baseline characteristics of cancer patients who were on fluoropyrimidine are shown in Table 1 (Characteristics of the patients). The mean (SD) age of the 50 patients who were followed up on fluoropyrimidine adverse events was 54 (12.75) years, and 30 (60%) were women. Colorectal cancer was the most common primary tumour site (26, 52%). The COVID-19 pandemic has forced healthcare systems to reorganise all activities to contain the virus infection. Therefore to minimise COVID-19 risk for a patient with cancer, oncology guidelines have recommended using the oral drug instead of intravenous (IV) preparation to minimise hospital admissions [25]. For this reason, most of the patients followed up were on capecitabine (88%), and only a few who couldn’t afford the oral drug or were otherwise prescribed by the physician based on performance status were on fluorouracil (12%).

### 3.2. Allele Frequency of DPYD Pathogenic Variants in the Guidelines

Deoxyribonucleic acid was prospectively obtained from 150 cancer patients and genotyped for *DPYD**2A, *13, c.2846A>T, and Hap 3B. The frequency of the four variants on the guidelines compared with other populations is shown in Table 2. None of the patients in the adverse events followed-up group and those that were eligible for fluoropyrimidine was a carrier for any of *DPYD* pathogenic variants (*2A, *13, c.2846A>T, or c.1679T>G), which are on the guidelines (CPIC, RNPGx, and DPWG). The *DPYD**13 is a rare variant in the population and was only observed in the European population (0.10%) according to the 1000 genome dataset. The *DPYD**2A was present across the populations, American (0.10%), European (0.45%), South Asian (0.34%), African (0.07%) and Zimbabwe (0.19%). Variant c.2846A>T is also a rare variant present across the general populations, American (0.30%), European (0.70%), South Asian (0.10%), African (0.08%) and Zimbabwe (0.29%). The frequency of the Hap B3 variant is high in the European (2.39%) and South Asian (1.90%) populations but low in other populations, American (0.60%) and African (0.08%) [15,26,27].

### 3.3. The Frequency of Fluoropyrimidine-Related AEs

The overall AEs data collected were stratified to capture severe fluoropyrimidine-related AEs during the early cycles (1–2) of fluoropyrimidine treatment. Of all the 50 patients recruited, only 27 were eligible for early fluoropyrimidine-related AEs. The overall outcome of the AEs is shown in the Appendix A, and Figure 3 shows the AEs markers. The histogram plot in Figure 2 shows that 33% of the accessed patients experienced severe AEs, and 67% experienced grade one or two AEs. None of the patients experienced severe gastrointestinal AEs. The major contributor to the severe global AEs was haematological. About 30% of the patients experienced grade one or two gastrointestinal AEs, and 52% experienced haematological AEs, as shown in the Appendix A. Only 2% of the patients experienced dose reduction due to severe AEs during the first two cycles, as also shown in the supplemental data. 

Fifty patients were assessed for severe (grade ≥ 3) AEs, and the median treatment cycle was three, ranging from cycle two to cycle six, as shown in the Appendix A. The histogram plot in Figure 3 shows that the frequency of global severe adverse events was 36%, and haematological AEs contributed 32% to the global AEs compared to gastrointestinal AEs, 4% and hand-foot-syndrome AEs, 2%. The frequency of grade ≤ 2 was 56%, also due to haematological AEs (50%), hand-foot syndrome (42%) and gastrointestinal AEs (20%), as shown in Figure 4. Six per cent (6%) and two per cent (2%) experienced dose reduction and treatment discontinuation during the total treatment period due to severe AEs, respectively, as shown in the Appendix A.

Severe fluoropyrimidine-related AEs data were obtained from the historical cohort [23], consisting of AEs data without genotype-guided dosing, and the literature cohort [24], consisting of AEs data with genotype-guided dosing as shown in the Appendix A. These data were compared with the present cohort, and the frequency of AEs in the histogram plot in Figure 5 shows that the severe AEs experienced in this cohort were higher (36%) than AEs experienced in the historical (33.6%) and literature (31%) cohort. There was no data for haematological and gastrointestinal severe AEs in the historical cohort. However, this cohort experienced more haematological AEs (32%) compared with the literature cohort (11.7%) and fewer gastrointestinal AEs (4%) compared with the literature cohort (12.5%).

### 3.4. Other Potential Risks of Developing AEs Using Gastrointestinal and Haematological Adverse Events as Surrogate Markers

Based on the literature, we examined the three major toxicity categories associated with fluoropyrimidines to find possible clinical predictors of the AEs. We employed a two-sample t-test on the BSA and BMI of patients who experienced severe AEs during their treatment period. Figure 6 shows that BSA (*p* = 0.0074) and BMI (*p* = 0.0001) were significantly associated with severe global (haematological, gastrointestinal, and hand-foot syndrome) AEs. The median BSA in patients with and without severe AEs was 1.61 m^2^ and 1.70 m^2,^ respectively, and the median BMI in patients with and without severe AEs was 26.7 kg/m^2^ and 22.5 kg/m^2,^ respectively.

We employed principal component analysis using k-means clustering to determine how well the test model fits our data. The k-means clustering in Figure 7A shows two distinct clusters, although with an overlap of PC1 and PC2. In Figure 7B, the red coloured circles indicate that the patient experienced severe fluoropyrimidine-related AEs at some point during their treatment.In contrast, the black circles indicate no severe fluoropyrimidine-related AEs. The clustering of the raw data before it was log-transformed showed that patients with BMI close to obesity grade 1 (BMI > 28 kg/m^2^) are more likely to experience severe AEs. In addition, patients with BSA > 1.52 m^2^ might experience severe adverse events, and the possibility of experiencing severe AEs increases with increasing BSA., as shown in Figure 7B. Patients within the area rectangle X in Figure 6B (BMI < 28 kg/m^2^ and BSA < 1.52 m^2^) did not experience AEs; however, all patients within the area rectangle Y (BMI > 28 kg/m^2^ and BSA > 1.52 m^2^) experienced severe AEs.

## 4. Discussion

This prospective observational study in patients on or scheduled to receive fluoropyrimidines showed that none of the 150 patients carried any of the four pathogenic *DPYD* variants. However, of the 50 patients on either the prodrug, capecitabine, or the 5-FU, 36% experienced severe AEs. The findings from the study indicate that current clinical pharmacogenetic guidelines based on the investigated *DPYD* variants with the high predictive value of early AEs in the European population were not observed in identifying patients at risk of AEs in this cohort.

Pharmacogenetic guidelines for fluoropyrimidine dose adjustment were drafted based on four *DPYD* variants (*DPYD**2A, *13, rs67376798, and rs75017182). The frequency of *DPYD**2A, *13, rs67376798, and rs75017182 in the European population is 0.45%, 0.10%, 0.70% and 2.4%, respectively. The overall prevalence of *DPYD* variant alleles is 3.65%, although it ranges from 3.46–7.0% in the literature [7,14,28]. In this focused study, none of the patients was a carrier of any of the variants. This result correlates with other published data where the allele frequency of the four variants round up to zero in the 1000 genome data [26].The *DPYD**2A, *13 and c.2846A > T round up to zero too in Zimbabwe’s general population [27]. In 2016, 2,038 patients were prospectively screened for *DPYD**2A in the Netherlands by Deenen and colleagues, of whom 22 (1.1%) were heterozygous polymorphic. It was a safety analysis study and the risk of grade 3 toxicity was thereby significantly reduced from 73% (95% CI, 58% to 85%) in historical controls (n = 48) to 28% (95% CI, 10% to 53%) by genotype-guided dosing; drug-induced death was reduced from 10% to 0%. It demonstrated for the first time that genotyping of *DPYD**2A is feasible and improves patients’ safety on fluoropyrimidine therapy [10]. A prospective safety analysis consisting of 1181 patients in 17 hospitals was carried out in the Netherlands by Henricks and colleagues in 2018. It further demonstrated the feasibility of prospective genotype-guided dosing of cancer patients being initiated on fluoropyrimidine. In the overall analysis, fluoropyrimidine-related severe toxicity was shown to be higher in *DPYD* variant carriers (33 [39%] of 85 patients) than in wild-type patients (231 [23%] of 1018 patients; *p* = 0.0013). Based on the findings, they concluded that implementing *DPYD* genotype-guided individualised dosing should be the new standard of care [13]. Some studies have been carried out to show the need for the incorporation of pretreatment *DPYD* testing into the standard of care for fluoropyrimidine regimens [3,7,29]. It is important to note that additional *DPYD* variants might play an important role in the African population, but these variants are not included in the guidelines [15,24]. The African-specific variants rs115232898-C have been shown to have a harmful or no function effect on DPD activity and was mentioned in the CPIC guideline [9], but neither dose reduction nor dose optimisation was provided in the guidelines [15,29]. This requires validation and may then be considered for inclusion in the guidelines for African populations. The rs61622928-T and rs2297595-C variants are also specific in African populations but require functional characterisation and conclusive interpretations [15,30]. Adding these (rs115232898-C, rs61622928-T, and rs2297595-C) African-specific variants might help improve the sensitivity of this testing and the safety outcomes of patients on fluoropyrimidine treatment [13]. Further functional and prospective genotyping studies should be explored in Africans and minority populations to understand the impact of their treatment outcome.

In early fluoropyrimidine treatment, the frequency of severe fluoropyrimidine AEs in this study (33%) is in discordance with some studies [5,14], where severe AEs amongst the *DPYD* wild-type group ranges were between 20% and 30%. Wigle and co-workers reported that the frequency of global AEs amongst patients without a variant carrier was 21%, and the haematological, gastrointestinal, and hand-foot syndrome were 7.6%, 9.7% and 1%, respectively [24]. Zimbabwe cohort’s, haematological AEs (33%) markers were the only contributor to early fluoropyrimidine-related severe AEs. Similar to the frequency of severe AEs as discussed earlier, the frequency of global grade ≤ 2 AEs was higher (67%) in this study cohort as compared to other studies, where the frequency of global grade ≤ 2 AEs was <52% amongst *DPYD* wild type carriers [13,28,31]. We further compared severe AEs in this cohort with AEs recorded in literature and historical cohorts. We obtained historical and literature values for global incidence risk of severe fluoropyrimidine-related AEs without genotype guidance from a meta-analysis and the implementation of a genotype-guided dosing study, respectively [23,24]. In comparison with these prior studies, the Zimbabwe cohort has a higher frequency of severe AEs (36%); the historical cohort without genotype-guided dosing was 34%, and literature cohorts with genotype-guided dosing had the lowest frequency of severe AEs (30%) [13,24]. Although the sensitivity of the present recognised four actionable variants (*DPYD**2A, *13, rs67376798, and rs75017182) for severe AEs is low. It only accounts for 30% of AEs in populations with the variants and might not be predict severe AEs in the African population. This is why overall severe AEs are still generally high (up to 30%) in the genotype-guided cohort [24] and higher (36%) in this study. 

The standard approach for personalising fluoropyrimidine dose is through body surface area, although neither flat dosing nor BSA personalised dosing is optimal for preventing adverse events in patients on fluoropyrimidine [32,33,34,35]. This study gave a flat dose of 3000 mg/day to patients with BSA between 1.32 m^2^ and 1.7 m^2^, while 4000 mg/day was given to patients with BSA between 1.75 m^2^ and 2.03m^2^. Association analysis using a two-sample t-test between BSA, BMI and severe global AEs showed a significant association between BSA and severe AEs (*p* < 0.05) and between BMI and severe AEs (*p* < 0.05). We further conducted a principal component analysis using k-means clustering to determine how well the test model fits our data. The two components in Figure 7A showed two different clusterings based on whether they experienced severe AEs, using BSA and BMI as the variables for clustering. Patients with high BSA and BMI that experienced severe fluoropyrimidine-related AEs clustered together, while patients with no severe AEs clustered separately. It agrees with other studies that have suggested an increased risk of having a worse outcome in obese cancer patients dosed based on BSA due to compromised pharmacokinetics parameters [34,36]. The weight or body surface area (BSA)-based approaches may fail to fully reflect the complexity of obesity in cancer patients [29]. Also, patients with reduced activity in the rate-limiting enzyme for dihydropyrimidine dehydrogenase are at high risk of supra-therapeutic drug concentrations under BSA-based standard dosing and consequently risk developing severe or sometimes even lethal fluoropyrimidine-related AEs [7,14]. Some patients with high BSA and experienced severe AEs might be variant carriers, considering that three African-specific variants that have implications on the *DPYD* gene were not tested for in this cohort. Another hypothesis is the complicated pharmacokinetics of fluoropyrimidine in obese patients. For example, fat deposits in obese patients slow the blood flow, which affects drug clearance and elimination [29]. It is also important to note that other environmental factors and performance status might contribute to high AEs in overweight cancer patients in this cohort. Also, interactions with other drugs were not explored as a possible cause of fluoropyrimidine-related AEs.

This study on the clinical utility of *DPYD* PGx testing is a precision medicine approach to assist physicians in tailoring fluoropyrimidine dosages; however, such studies can be challenging in our setting. One of the significant limitations of this study is the small sample size (n = 150) in this population, and only a few patients might be detected if African-specific variants were included. Also, another important limitation is that the African-specific variant c.557A>G (rs115232898, p.Y186C), established to decrease the function of DPD activity, was not included based on the aim of the study. The objective is to focus on the available pathogenic variants with treatment optimization or dose reduction in the guideline. The under-resourced clinical healthcare systems are one of the significant challenges. For example, electronic health records (EHRs) are critical to obtaining longitudinal phenotype and genotype patient data for effective patient management, but this was lacking in our system. Also, co-infection and co-morbidities patterns are not uniform across the recruited patients. It constitutes a significant challenge for disease management because treatment regimens and ADR patterns differ across patients. Polypharmacy also complicates AEs assessment because of the patients’ prescribed multidrug regimens. The immediate shift in cancer patient management from i.v. to oral administration also affected ‘patients’ recruitment to the study and AEs monitoring. Therapeutic drug monitoring (TDM) in patients that received i.v. 5-FU treatment was lacking, making it difficult to confirm drug concentration values, especially in patients that experienced severe AEs. Future studies on the implementation or utility of clinical PGx testing in Africa will be more feasible by incorporating patient data into EHRs. Optimal TDM should also complement genotype and phenotype data to interpret observed AEs adequately, as suggested in other studies [29,33,37].

## 5. Conclusions

This is the first study to evaluate the potential clinical utility of *DPYD* pharmacogenetic clinical guidelines in a cohort of cancer patients of African ancestry. The provisional results show the potential limitation of the current PGX guidelines based on the four *DPYD* variants to identify patients at risk of AEs in this cohort. Although the four variants used to draft the current guidelines were not predictive of severe AEs in this cohort and seemed more predictive of severe AEs in patients of western descent. There is a need to expand on African data to inform guidelines of fluoropyrimidine drug safety. Therefore, a significant multicenter study, cutting across different population-specific variants, will help generalise the supporting guidelines for preventing fluoropyrimidine-related adverse events.

## Figures and Tables

**Figure 1 jpm-13-00588-f001:**
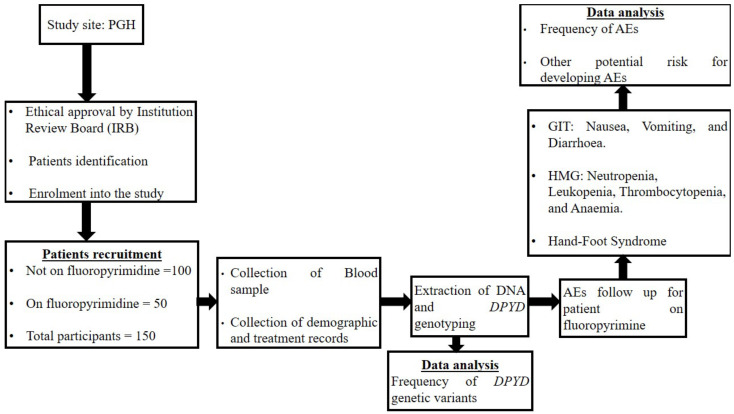
Flow diagram illustrating the visual representation of the methodology (AEs., adverse events; *DPYD.*, dihydropyrimidine dehydrogenase; GIT., gastrointestinal; HMG., haematological; P.G.H., Parirenyatwa Group of Hospitals).

**Figure 2 jpm-13-00588-f002:**
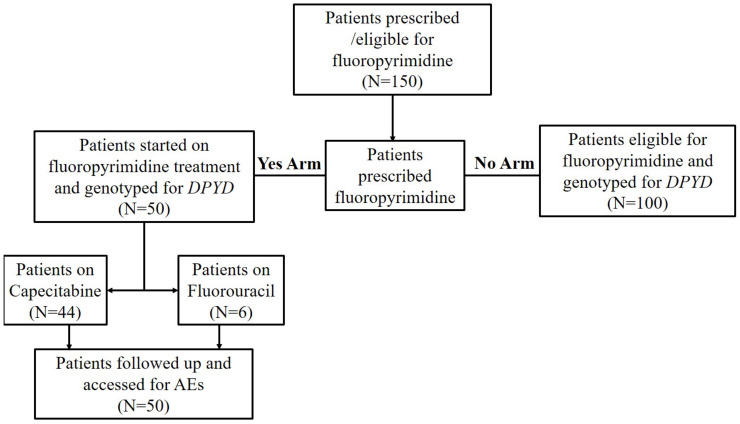
Flow diagram illustrating the study cohort (AEs., adverse events; *DPYD.*, dihydropyrimidine dehydrogenase).

**Figure 3 jpm-13-00588-f003:**
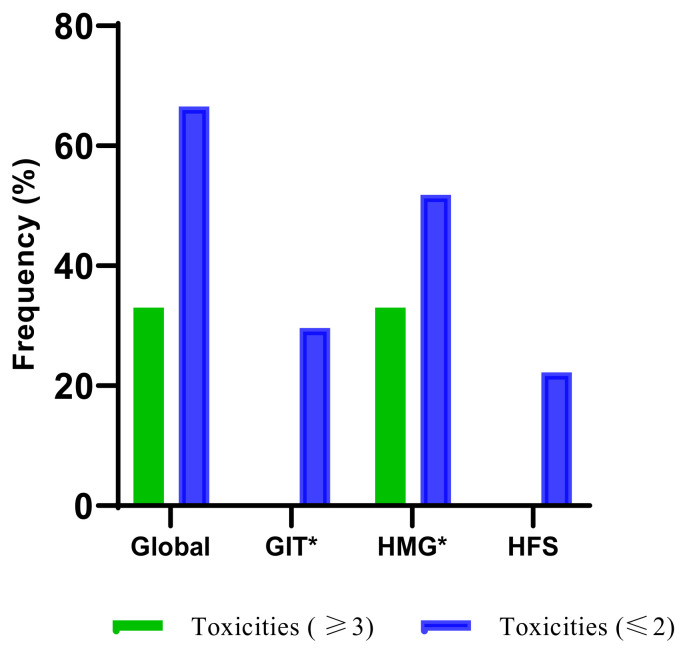
Plot of early fluoropyrimidine adverse events markers (GIT., HMG., HFS.) during the first two treatment cycles. (GIT., gastrointestinal; HMG., haematological; HFS., hand-foot-syndrome). Global includes all fluoropyrimidine-related AEs. GIT* Gastrointestinal markers include nausea, vomiting, and diarrhoea. HMG* Haematological markers include neutropenia, leukopenia, thrombocytopenia, and anaemia. In grades ≥ 3, this also includes dose reduction and treatment discontinuation. In grade ≤ 2, this does not include dose reduction and treatment discontinuation.

**Figure 4 jpm-13-00588-f004:**
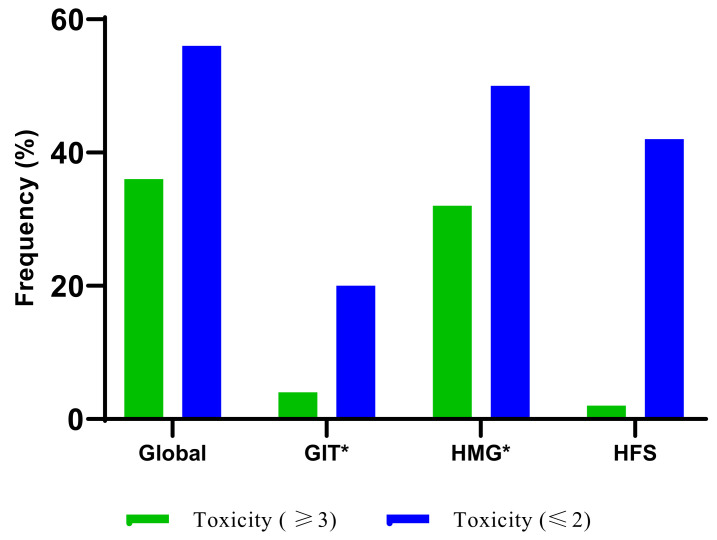
Plot of fluoropyrimidine adverse events markers during the total study period. (GIT, gastrointestinal; HMG, haematological; HFS, hand-foot-syndrome). Global includes all fluoropyrimidine-related AEs. GIT* Gastrointestinal markers include nausea, vomiting, and diarrhoea. HMG* Haematological markers include neutropenia, leukopenia, thrombocytopenia, and anaemia. In grades ≥ 3, this also includes dose reduction and treatment discontinuation. In grade ≤ 2, this does not include dose reduction and treatment discontinuation.

**Figure 5 jpm-13-00588-f005:**
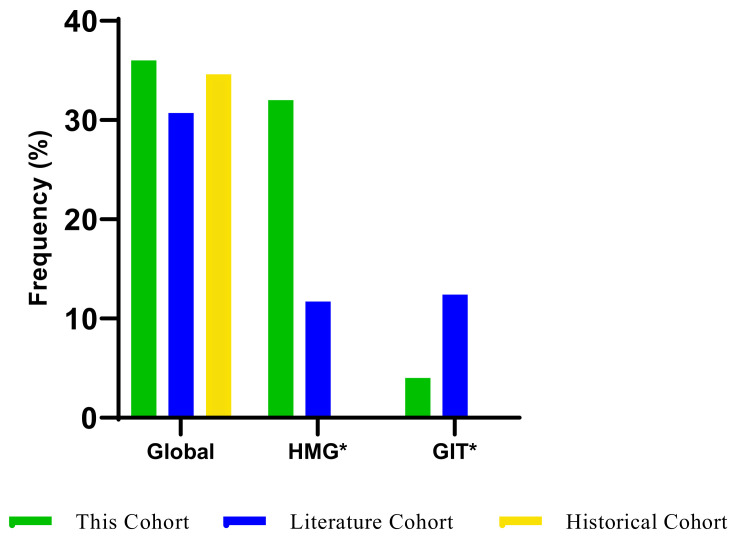
Plot of global and overall adverse events markers in 3 cohorts. Global includes all fluoropyrimidine-related AEs grade ≥ 3. It might not include dose reduction and treatment discontinuation. GIT* Gastrointestinal markers include nausea, vomiting, and diarrhoea. HIG* Haematological markers include neutropenia, leukopenia, thrombocytopenia, and anaemia.

**Figure 6 jpm-13-00588-f006:**
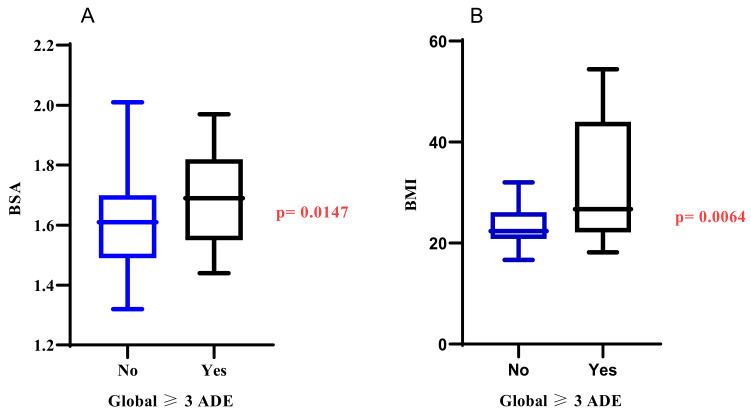
Association between BSA (**A**), BMI (**B**) and severe fluoropyrimidine-related adverse events markers. BSA., Body Surface Area; BMI., Body Mass Index; ADE., Adverse Drug Events; Global ≥ 3., Global severe adverse events. No., no severe global AEs. Yes., there are severe global AEs.

**Figure 7 jpm-13-00588-f007:**
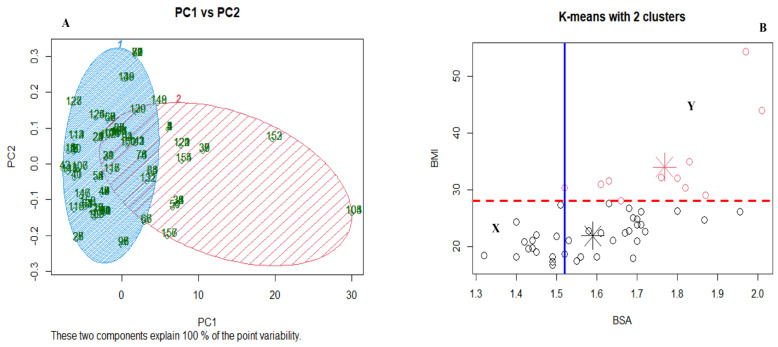
Principal component analysis with BSA and BMI k-means clustering using global severe adverse events as the confusion matrix. Principal Component 1 vs Principal Component 2 (**A**). Clustering of the cohort based on adverse events using k-means cluster (**B**). PC1., Principal Component 1; PC2., Principal Component 2; BMI., Body Mass Index; BSA., Body Surface Area.

**Table 1 jpm-13-00588-t001:** Characteristics of the patients.

Characteristics	N = 50 (%)
Age. mean (SD)	54.08 (12.75)
Sex Female Male	30 (60) 20 (40)
BSA, Median (Q1, Q3)	1.63 (1.49, 1.71)
BMI, Median (Q1, Q3)	22.7 (19.8, 27.6)
HIV Status Positive Negative	11 (22) 39 (78)
Tumour site N (%) Colorectal Breast Gastric and oesophagus Cervical Pancreas Others (Head & neck and liver)	26 (52) 9 (18) 8 (16) 3 (6) 2 (4) 2(4)
Chemotherapy regimen CAPEOX CAPE 5-FU + Cisplatin CAPE + Cisplatin CAPE + GEM 5-FU + Others ^a^	29 (58) 12 (24) 3 (6) 1 (2) 2 (4) 3 (6)
AJCC Group Staging I II III IV	2 (4) 11 (22) 14 (28) 23 (46)

AJCC., American Joint Committee on Cancer; CAPEOX., Capecitabine & Oxaliplatin; CAPE., Capecitabine; 5-FU., 5-Fluorouracil; GEM., Gemcitabine; HIV., Human Immunodeficiency Virus; BSA., Body Surface Area; BMI., Body Mass Index; SD., Standard Deviation. ^a^ Others include Leucovorin, Carboplatin, Methotrexate, cyclophosphamide.

**Table 2 jpm-13-00588-t002:** The frequency of CPIC *DPYD* pathogenic variants in different populations.

Variant	RsID	Nucleotide Change	Single AA Change	DPD Activity	America ^a^	Europe ^a^	SAS ^a^	Africa ^a^	Zimbabwe ^b^	This Cohort
*2A	rs3918290	c.190511G>A	Not changed	No activity	0.0010	0.0045	0.0034	0.0007	0.0019	0.0000
*13	rs55886062	c.1679T>G	p.I560S	No activity	0.0000	0.0010	0.0000	0.0000	0.0000	0.0000
NA	rs67376798	c.2846A>T	p.D949V	Decreased activity	0.0030	0.0070	0.0010	0.0008	0.0029	0.0000
Hap B3	rs75017182	c.1129-5923C>G	Not changed	Decreased activity	0.0060	0.0239	0.0190	0.0008	NA	0.0000
Type of population	NA	NA	NA	NA	General population	General population	General population	General population	General population	Focused Population
Population size	NA	NA	NA	NA	694	1006	978	1322	522	150

Abbreviations: NA., not applicable; Hap B3., haplotype- B3; RsID., reference SNP identification; A.A., amino acid; DPD., dihydropyrimidine dehydrogenase; SAS., South Asian. ^a^ [26] ^b^ [27].

## Data Availability

Data availability is available on request.

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
