# Peer review of "Pharmacogenetics and Adverse Events in the Use of Fluoropyrimidine in a Cohort of Cancer Patients on Standard of Care Treatment in Zimbabwe"

_jpm, 2023, doi:10.3390/jpm13040588_

Round 1

Reviewer 1 Report

This manuscript addresses the potential usefulness of fluoropyrimidine pharmacogenetic clinical guideline based on DPYD genotyping (4 variants) in a Zimbabwean patient cohort. In addition, it explores the occurrence of severe adverse effects related to anthropometric factors in this cohort. Authors conclude that adverse effects are common in the analysed population despite the absence of the variants of DPYD suggested by clinical guideline. Two factors (BSA and BMI) are revealed as relevant for the presence of adverse effects.

Although there is some controversial about the impact of implementation of pharmacogenetics in relation to fluoropyrimidine therapy, in this paper ethnicity is pointed out as one important factor to take into account.

I include some suggestions for the improvement of the paper:

1)      Lines 48-51: the sentence should be rewritten.

2)      Methods. DNA extraction procedure is not specified.

3)      English should be checked.

4)      Initial capital/small letter for drugs should be maintained.

5)      Table 1. Spelling mistake about cyclophosphamide.

6)      Identification of the different polymorphisms of DPYD gene should be maintained (rs code/*number/nucleotide+position variation) throughout the paper or include an initial explanation about the different identification names for each variant (table 2).

7)      Importance and differences between the guidelines (CPIC, DPWG, FRSWN) could be added.

8)      Line 203. Is it figure 3 or 4?

9)      Figure 5. Letter size should be homogeneous. Data numbers are not shown in the figure.

10)   Figure 6. What does it mean red/black coloured circles? It should be explained in order to clarify it.

11)   Line 302. “in this cohort” should be added.

12)   Lines 360-362. What would be consequences of carrying these SNPs? Lack/decrease/increase of function of the enzyme encoded by DPYD gene should be added for each variant for a better understanding of its presence in patients.

13)   Interactions with other drugs have not been explored as a possible cause of occurrence of adverse drug reactions related to fluoropyrimidines.

Reviewer 2 Report

General Comments:

The manuscript entitled " Pharmacogenetics and Adverse Events in the use of Fluoropy- 2 rimidine in a cohort of cancer patients on standard of care treat- 3 ment in Zimbabwe" is comprehensive review of  the current advances in the field PGx.

After reviewing this manuscript, I do have some concerns about the research.

1.     Please used italic once author would reveal the genes

2.     Lines 21-22: Author revealed that “The frequency of severe AEs was however high (36%) and comparable to frequencies reported in literature. Is the frewquency allele that author claimed same with the prevalence of AE in Zimbabwe?

3.     Line 64-66: please site the reference related to the sentences.

4.     Author should strongly elaborate the gap why authors interested to doing the Pharmacogenetics and Adverse Events in the use of Fluoropy- 2 rimidine in a cohort of cancer patients on standard of care treat- 3 ment in Zimbabwe?

5.     Eventhough author revealed that “There are no data on the PGx and pharmacokinetics (PK) of fluoropyrimidines or the prevalence of the fluoropyrimidines related adverse events based on clinical evidence in people of African ancestry”. Author should explain the prevalence of the patients treated by the fluoropyrimidines.

6.     Number of ethical clearance should be provided

7.     I highly recommended the methodology part should be described using Figure/visualization.

8.     The frequency of CPIC DPYD pathogenic variants in different populations should be explained in detail for every frequency in multiple continents.

9.     I think author also can use the gene expression information using bioinformatic database to link the variants and the gene expression in such tissues.

Round 2

Reviewer 2 Report

My concer regarding the manuscript is the copyright issue of Figure 1. Just to make it ascertain wether author have provided the license of figure. beside, the resolution is too low. 

Author does not response my suggestion regarding my comment nomer 9.

 I think author can use publicly available database such as GTEx etc to evalute the expression of variants in each tissue.

Author Response

Dear Reviewer,

Thank you for your comments regarding the revised manuscript. You have highlighted very important points that should be addressed, which we have responded to.

  1. We designed Figure 1 using Microsoft power point and then converted it to an image while still retaining its highest quality in Microsoft. The ideas incorporated in the figure's design that might result in copyright infringement have been removed. However, it still clearly shows the visual representation of the methodology

Figure 1: Flow diagram illustrating the visual representation of the methodology design (AEs, adverse events; DPYD, dihydropyrimidine dehydrogenase; P.G.H., Parirenyatwa Group of Hospitals)

  1. We responded to comment 9 initially from your first review that it is a good idea to look at the expression levels of the DPYD gene, especially in patients with obvious phenotypic traits (severe AEs) in our cohort. However, looking at the expression level of the gene in this study is beyond our scope.

Most databases have other information on the DPYD variant SNPs. Still, they do not have the expression levels of each variant, although they have the normal expression level of the DPYD, including GTEx (https://www.gtexportal.org/home/gene/DPYD) and on NCBI (https://www.ncbi.nlm.nih.gov/gene/1806). Hands-on, these databases will show that the DPYD gene is highly expressed in the liver and blood tissue, especially the white blood cell, which is our knowledge.

However, from the literature, some studies have shown the expression levels of DPYD variants in tissues, especially the blood tissue. A good example is a paper by offer et al. (https://doi.org/10.1158%2F0008-5472.CAN-13-2482) showing the expression levels of DPYD variants compared to normal.

We know that these variants significantly reduce the enzyme activity produced by the DPYD gene, which affects the rate at which it metabolizes fluoropyrimidine. Therefore, looking at the expression levels of the DPYD is a good idea, but it wasn't the aim and was beyond the study's scope.

Thank you for your feedback.
